# Mechanically active integrins target lytic secretion at the immune synapse to facilitate cellular cytotoxicity

Mitchell S. Wang[1,2], Yuesong Hu[3], Elisa E. Sanchez[1,4], Xihe Xie [5], Nathan H. Roy[6], Miguel de Jesus [1], Benjamin Y. Winer[1], Elizabeth A. Zale[1], Weiyang Jin[7], Chirag Sachar[7], Joanne H. Lee[7], Yeonsun Hong[8], Minsoo Kim [8], Lance C. Kam[7], Khalid Salaita [3] & Morgan Huse [1✉]

Cytotoxic lymphocytes fight pathogens and cancer by forming immune synapses with infected or transformed target cells and then secreting cytotoxic perforin and granzyme into the synaptic space, with potent and specific killing achieved by this focused delivery. The mechanisms that establish the precise location of secretory events, however, remain poorly understood. Here we use single cell biophysical measurements, micropatterning, and functional assays to demonstrate that localized mechanotransduction helps define the position of secretory events within the synapse. Ligand-bound integrins, predominantly the $\alpha_L\beta_2$ isoform LFA-1, function as spatial cues to attract lytic granules containing perforin and granzyme and induce their fusion with the plasma membrane for content release. LFA-1 is subjected to pulling forces within secretory domains, and disruption of these forces via depletion of the adaptor molecule talin abrogates cytotoxicity. We thus conclude that lymphocytes employ an integrin-dependent mechanical checkpoint to enhance their cytotoxic power and fidelity.

[1] Immunology Program, Memorial Sloan Kettering Cancer Center, New York, NY, USA. [2] Pharmacology Program, Weill-Cornell Graduate School of Medical Sciences, New York, NY, USA. [3] Department of Chemistry, Emory University, Atlanta, GA, USA. [4] Biochemistry and Molecular Biology Program, Weill-Cornell Graduate School of Medical Sciences, New York, NY, USA. [5] Neuroscience Program, Weill-Cornell Graduate School of Medical Sciences, New York, NY, USA. [6] Department of Microbiology and Immunology, State University of New York Upstate Medical University, Syracuse, NY, USA. [7] Department of Biomedical Engineering, Columbia University, New York, NY, USA. [8] Department of Microbiology and Immunology, University of Rochester Medical Center, Rochester, NY, USA. ✉email: husem@mskcc.org

The secretory output of cell–cell interfaces must be tightly controlled in space and time to ensure functional efficacy. This is particularly true for cytotoxic lymphocytes (cytotoxic T lymphocytes (CTLs) and natural killer (NK) cells) which release a toxic mixture of granzyme proteases and the pore-forming protein perforin to destroy infected or transformed target cells[1]. Perforin disrupts the plasma membrane, enabling granzymes to access the target cell cytoplasm, where they cleave multiple substrates to induce cytoskeletal collapse, mitochondrial dysfunction, and programmed cell death[2,3]. Essentially all eukaryotic cell types are vulnerable to this killing mechanism. Hence, to ensure the safety of innocent bystander cells in the surrounding milieu, cytotoxic lymphocytes must release perforin and granzyme so that only the target membrane is accessible. Precisely how they do this remains a topic of intensive research.

Mature perforin and granzyme are stored in specialized secretory lysosomes called lytic granules. Target cell recognition, typically mediated by the T-cell antigen receptor (TCR) on CTLs or by activating NK receptors on NK cells, mobilizes these granules, which move toward the immune synapse (IS) and fuse with the synaptic membrane, releasing their contents into the intercellular space[4]. This directional trafficking behavior has long been attributed to the lymphocyte centrosome, which polarizes to a position just beneath the center of the IS within minutes of target cell engagement[4,5]. Because lytic granules cluster around the centrosome, its reorientation places them in close apposition with the synaptic membrane, where they are presumably well-positioned for fusion. Although a number of studies support a role for the centrosome in granule delivery to the IS[6–10], we and others have found that proper centrosome reorientation is not strictly required for synaptic secretion[11–14]. Indeed, even complete depolymerization of the microtubule cytoskeleton fails to alter the directionality of the secretory response[14]. Hence, other mechanisms must control where lytic granules dock and fuse.

Cytotoxic synapses are mechanically active structures capable of exerting nanonewton scale forces against the target cell[15]. These forces have been implicated in both the activation of mechanosensitive cell-surface receptors on the lymphocyte and the subsequent potentiation of perforin activity[16–22]. We have found that lytic granule exocytosis (also called degranulation) tends to occur in regions of active force exertion within the IS[16], raising the possibility that local mechanosensing might play an instructive role in guiding perforin and granzyme release. This hypothesis is particularly intriguing because several receptors with established roles in IS formation, including the TCR, multiple activating NK receptors, and the integrin LFA-1 (for lymphocyte function-associated antigen 1), are known to be mechanosensitive[22–25].

Of these proteins, LFA-1 stands out for being especially attuned to detecting and delivering forces. As with other integrins, LFA-1 occupies multiple conformational states[26], each of which exhibits a distinct affinity for its ligands, the intracellular

adhesion molecules (ICAM) 1 and 2. In its resting, bent conformer, the ligand-binding activity of LFA-1 is weak. "Inside-out" signals, typically driven by activating receptors like the TCR, induce LFA-1 extension into an intermediate affinity state[27,28]. Optimal ligand binding, however, requires that LFA-1 be coupled to the cortical filamentous actin (F-actin) cytoskeleton[29–31]. This places the integrin under tension, enabling both the formation of a high affinity "catch-bond" with ICAM and the assembly of "outside-in" signaling complexes around its tail domains[32]. In this manner, LFA-1 establishes strong adhesion to the target cell while concomitantly generating costimulatory signals that boost lymphocyte activation[33–36]. Importantly, prior studies also indicate that LFA-1 engagement promotes the polarization of lytic granules to the IS and their accumulation in synaptic domains containing the extended, F-actin bound form of the integrin[37–39]. The extent to which ligand-bound LFA-1 actually dictates the site(s) of degranulation, however, is unclear.

In this study, we combine micropatterning, DNA-based tension sensors, and functional readouts of cytotoxicity to investigate the mechanisms guiding granule release at the IS. Our results indicate that integrins like LFA-1 function as mechanical gates for degranulation, coupling cytotoxic secretion to the capacity of a lymphocyte to form a close, physically active contact with its target. This mechanism likely contributes to both the potency and the specificity of target cell killing.

## Results

### LFA-1 engagement is required for robust degranulation.

Murine CD8[+] CTLs express high levels of LFA-1 and are potent killers, making them an excellent model system for studying the interplay between integrin engagement and cytotoxicity. For most of our studies, we used CTLs expressing the OT-1 TCR, which binds to the ovalbumin$_{257-264}$ peptide (OVA) in the context of the class I MHC protein H-2K$^b$. Recognition of this cognate peptide-MHC (pMHC) induces the migrational arrest of OT-1 CTLs and the formation of mechanically active synapses. The pattern and magnitude of forces within these contacts can be visualized by micropillar-based traction force microscopy[40]. Using this approach, we found that surfaces containing pMHC and ICAM-1 elicited substantially greater synaptic force exertion than did surfaces containing pMHC alone (Fig. 1). This result implied that LFA-1 is a key player in IS mechanics and therefore a candidate for mediating coupling between mechanical and secretory output.

Next, we used a neutralizing antibody that specifically disrupts LFA-1-ICAM-1/2 binding to evaluate the importance of this interaction for cytotoxic secretion. LFA-1 blockade strongly suppressed target cell lysis in cocultures of OT-1 CTLs and OVA-loaded RMA-s targets (Fig. 2a, b and Supplementary Figs. 1a and 2a). This cytotoxicity defect was associated with markedly reduced degranulation, which we measured by surface exposure of the lysosomal marker Lamp1 (Fig. 2c and Supplementary Figs. 1b, 2b, 3a)

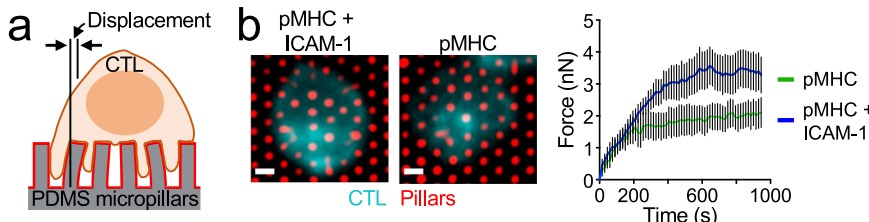

**Fig. 1 LFA-1 engagement is required for synaptic force exertion. a** OT-1 CTLs labeled with fluorescent anti-CD45 Fab were stimulated on PDMS micropillar arrays coated with pMHC ± ICAM-1. Traction forces were derived from pillar displacement. **b** Left, representative images of CTLs interacting with the arrays. Scale bars = 2 μm. Right, mean force exertion against the array graphed against time. N ≥ 8 cells for each sample. Error bars signify SEM. Data are representative of two independent experiments. Source data are provided as a Source Data file.

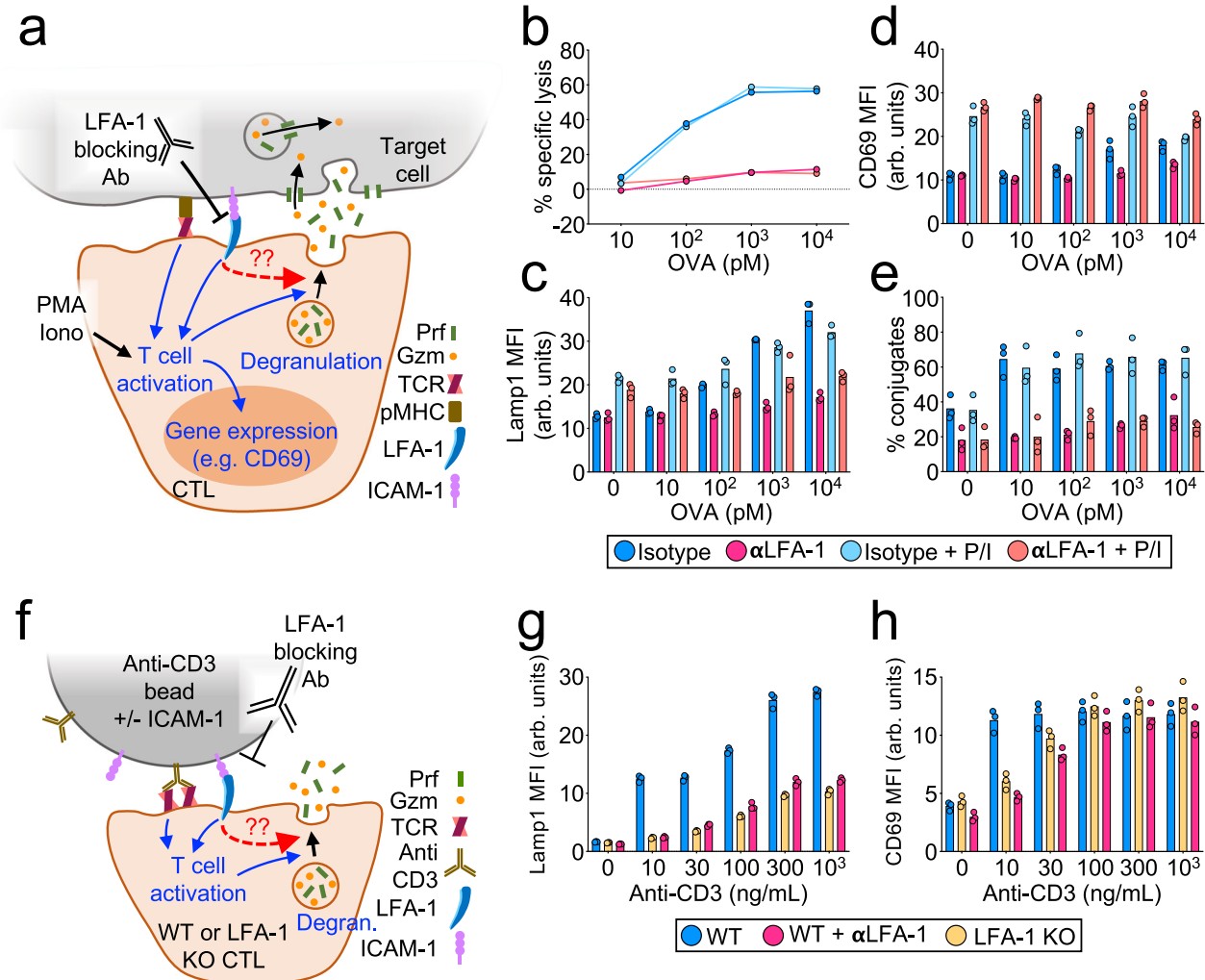

**Fig. 2 LFA-1 is required for CTL degranulation and cytotoxicity. a–e** OVA-loaded RMA-s target cells were mixed with OT-1 CTLs in the presence of LFA1-blocking antibody (αLFA-1) or isotype control. PMA/Iono was applied to some samples in order to drive TCR-independent CTL activation. **a** Schematic diagram of the experimental design. **b** Target cell killing, measured 4 h after CTL-target cell mixing ($N = 2$ replicate experiments). **c** Lamp1 exposure (degranulation), measured 90 min after CTL-target cell mixing ($N = 4$ replicate experiments). **d** CD69 expression, measured 90 min after CTL-target cell mixing ($N = 2$ replicate experiments). **e** Conjugate formation, measured 90 min after CTL-target cell mixing ($N = 3$ replicate experiments). **f–h** Polyclonal WT and LFA-1 KO CTLs were mixed with stimulatory beads coated with anti-CD3 antibody + ICAM-1 in the presence or absence of LFA-1-blocking antibody (αLFA-1) as indicated. **f** Schematic diagram of the experimental design. **g** Lamp1 exposure, measured 90 min after CTL-bead mixing ($N = 2$ replicate experiments). **h** CD69 expression, measured 90 min after CTL-bead mixing ($N = 2$ replicate experiments). Data points (**d**, **e**, **g**, **h**) represent technical triplicate measurements from an individual experiment. Data points in b represent mean values calculated from technical triplicates. Source data are provided as a Source Data file. Replicate studies are shown in Supplementary Fig. 2.

and by depletion of intracellular granzyme B (Supplementary Fig. 3b). Anti-LFA-1 also inhibited degranulation by human cytomegalovirus-specific CTLs cocultured with antigen-bearing targets (Supplementary Fig. 3c). To drill down on the molecular determinants of this cytotoxicity defect, we developed a reductionist system in which OT-1 CTLs were activated by beads coated with pMHC and/or ICAM-1 (Supplementary Fig. 4a). Beads containing both proteins (pMHC + ICAM-1) elicited significantly stronger degranulation than beads containing pMHC alone (Supplementary Fig. 4b), and the responses induced by the pMHC + ICAM-1 beads were inhibited by LFA-1 blockade (Supplementary Fig. 4c). To further validate the importance of LFA-1, we stimulated CTLs derived from *cd11a*−/− mice (LFA-1 KO) with beads containing anti-CD3 antibodies (to engage the TCR) either with or without ICAM-1 (Fig. 2f). Anti-CD3 was used instead of cognate pMHC because the LFA-1 KO mice were polyclonal. Degranulation was profoundly inhibited by LFA-1 deficiency,

essentially phenocopying the effects of LFA-1 neutralizing antibodies on wild-type cells (Fig. 2g and Supplementary Fig. 2e).

Collectively, these results were consistent with a specific role for LFA-1 in degranulation, but they did not rule out the possibility that LFA-1 engagement might promote this response secondarily by augmenting antigen-induced T-cell activation (Fig. 2a). To explore this possibility, we surveyed several indices of T-cell activation and found that LFA-1 blockade altered some, but not all, of these readouts. Anti-LFA treatment had no effect on antigen-induced proliferation (Supplementary Figs. 1c and 5a), which we assessed by dilution of CellTrace Violet dye. Similarly, we observed no changes in MAP kinase (MAPK) and PI-3 kinase (PI3K) signaling, which we measured by phosphorylation of Erk1/2 and Akt, respectively (Supplementary Figs. 5b and 12). By contrast, anti-LFA-1 antibody treatment dampened cytosolic calcium ($Ca^{2+}$) influx (Supplementary Fig. 5c), and

pharmacologic or genetic blockade of LFA-1 inhibited the upregulation of CD69, an immediate-early response gene (Fig. 2d, h and Supplementary Figs. 1b, 2c, 2f, 4d). To decouple these LFA-1-dependent effects on T-cell activation from a distinct and specific role in degranulation, we used a combination of phorbol myristate acetate (PMA) and the $Ca^{2+}$ ionophore A23187 (Iono) to induce T-cell activation in the absence of TCR engagement (Fig. 2a and Supplementary Fig. 4a). CTLs treated with PMA/Iono alone exhibited robust CD69 expression (Fig. 2d and Supplementary Figs. 2c, 4d), indicative of strong activation. Their degranulation responses were quite modest, however (Fig. 2c and Supplementary Figs. 2b, 4c), pointing to the importance of target contact/proximity in stimulating cytotoxic secretion. Indeed, robust Lamp1 exposure was only observed in CTLs that were concomitantly exposed to antigenic pMHC and ICAM, either on target cells or on beads. Critically, this ligand-induced component of degranulation was completely inhibited by LFA-1 blockade (Fig. 2c and Supplementary Figs. 2b and 4c). Taken together, these results indicate that LFA-1 engagement promotes cytotoxic secretion independently of T-cell activation and at a level downstream of early signaling events. Notably, LFA-1 was also required for the formation of strong CTL-target cell conjugates, both in the presence and in the absence of PMA/Iono (Fig. 2e and Supplementary Figs. 1d and 2d). Hence, the capacity of LFA-1 to stimulate degranulation was phenotypically linked to its ability to mediate strong adhesion.

**Degranulation occurs in regions of TCR and LFA-1 coengagement**. The observation that both the TCR and LFA-1 were necessary for robust degranulation (Fig. 2 and Supplementary Figs. 2–4) raised the possibility that both receptor types must be engaged within the same cell–cell interface to elicit directional secretory responses. This requirement would presumably enhance the specificity of killing by ensuring that toxic factors are released only against bona fide target cells (expressing cognate pMHC) that are tightly associated with the CTL (via integrin adhesion). To test this idea, we stimulated CTLs using beads coated with both pMHC and ICAM-1 (cis) or using a mixture of beads coated separately with only pMHC or ICAM-1 (trans) (Fig. 3a). Care was taken to make sure that the total amount of accessible pMHC and ICAM-1 was identical in each experimental group and that CTLs could engage multiple beads simultaneously (Supplementary Fig. 6a). Strikingly, immobilized ICAM-1 boosted TCR-

induced degranulation only when presented in the cis configuration (Fig. 3b and Supplementary Fig. 6b). Indeed, CTLs stimulated with the trans mixture of pMHC- and ICAM-1-coated beads responded similarly to CTLs stimulated with pMHC-coated beads alone. We observed the same pattern of results using CD69 upregulation as the downstream readout (Fig. 3c and Supplementary Fig. 6b). Hence, ligand-bound LFA-1 must share the same interface as the ligand-bound TCR in order to boost cytotoxicity and T-cell activation.

Building upon this idea, we next examined whether local coengagement of LFA-1 and the TCR could direct the position of degranulation events within the IS. Using protein microstamping[41], we prepared glass coverslips containing 2-μm spots of fluorescent streptavidin spaced in a 10 × 10-μm square grid. These surfaces were then incubated with mixtures of unbiotinylated proteins to coat/block the empty glass between streptavidin spots, and then with biotinylated proteins to load the spots themselves. By varying the composition of coating and loading mixtures, we were able to generate a panel of distinct micropatterned substrates: (1) ICAM-1 spots within a uniform background of pMHC (ICAM-spot), (2) pMHC spots on an ICAM-1 background (Antigen-spot), and (3) spots containing both ICAM-1 and pMHC on a nonstimulatory (BSA-coated) background (Dual-spot) (Fig. 4a). We also generated control surfaces containing empty streptavidin spots in a background of admixed pMHC and ICAM-1. CTLs plated on Dual-spot surfaces evinced $Ca^{2+}$ flux only during periods of spot contact (Supplementary Fig. 6c and Supplementary Movies 1 and 2), confirming that biotinylated stimulatory ligands could be constrained in space by the patterned streptavidin. To monitor the degranulation position in this system, we employed a reporter construct in which the pH-sensitive fluorescent protein pHluorin is linked to the lytic granule resident protein Lamp1[42]. Because pHluorin is quenched in the low pH granule environment, it becomes visible only upon fusion, generating a transient burst of fluorescence that reveals the position of the degranulation site within the IS. When CTLs expressing pHluorin-Lamp1 were imaged on ICAM-spot, Antigen-spot, or Dual-spot surfaces, degranulation events tended to cluster around the fluorescent spots containing ICAM-1 and/or pMHC (Fig. 4b, c and Supplementary Movie 3). In all three cases, the mean distance between degranulation events and the spots closest to them was substantially lower than one would expect by chance (dotted line in Fig. 4c) and significantly less than the corresponding distances measured between empty SA spots and degranulations on control surfaces. The enrichment of degranulation

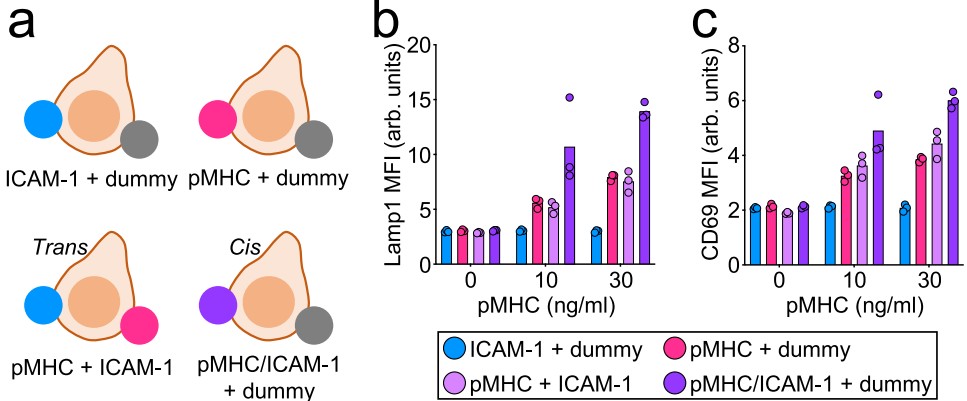

**Fig. 3 LFA-1 engagement promotes TCR-induced degranulation only within the same interface. a–c** OT-1 CTLs were activated by equal mixtures of stimulatory beads bearing the indicated proteins. **a** A schematic of the experiment. Dummy denotes beads coated with nonstimulatory pMHC (H-2D^b-KAVY) alone. Trans and cis co-presentation of H-2K^b-OVA and ICAM-1 are indicated. **b, c** Graphs showing CTL degranulation (**b**) and CD69 expression (**c**), measured 90 min after CTL stimulation (N = 2 replicate experiments). Data points represent technical triplicate measurements from an individual experiment. Source data are provided as a Source Data file. Replicate studies are shown in Supplementary Fig. 6.

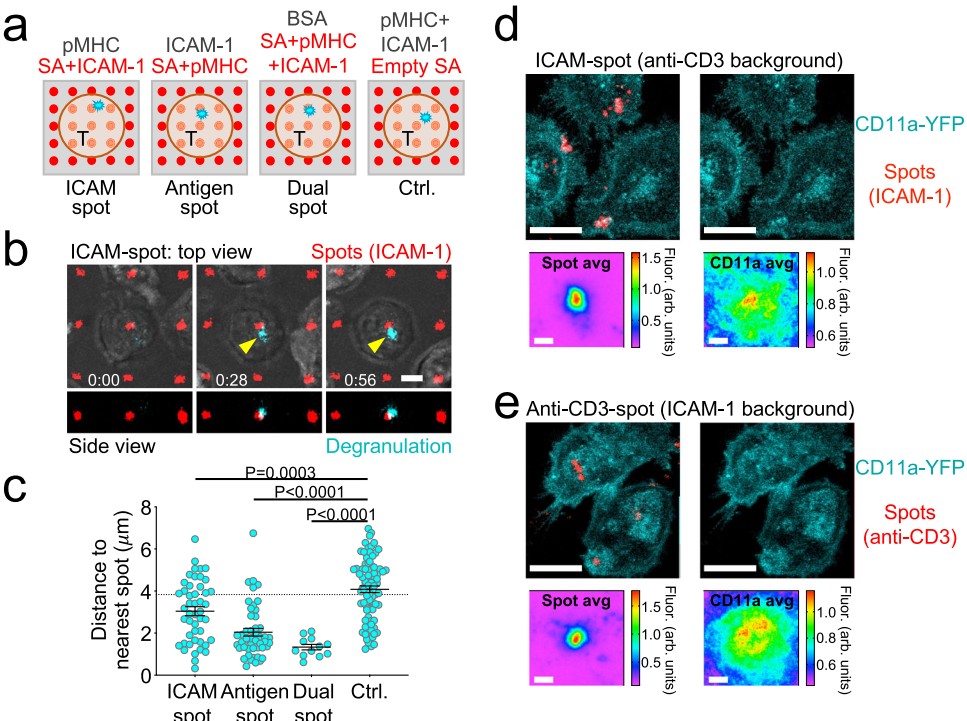

**Fig. 4 Degranulation occurs in IS domains containing both ligand-bound TCR and ligand-bound LFA-1. a–c** OT-1 CTLs expressing pHluorin-Lamp1 were imaged on micropatterned surfaces coated with stimulatory pMHC (H-2K$^b$-OVA) and ICAM-1 in four different configurations. **a** Schematic showing the configurations. SA streptavidin. **b** Time-lapse montage showing a representative degranulation event (indicated by the yellow arrowhead) on an ICAM-spot surface. In the top views, fluorescent signals have been superimposed on the brightfield image. Time in M:SS is shown at the bottom left corner of the top view images. Scale bars = 4 μm. **c** Distance between each degranulation event and the closest fluorescent SA spot in the IS. The dotted line indicates the mean distance expected from randomly placed degranulation. $N = 12$ for Dual-spot and $\geq 47$ for the other conditions. Error bars signify SEM. $P$ values were calculated by one-way ANOVA with Tukey correction. **d, e** CD11a-YFP CTLs were imaged on ICAM-spot (**d**) and Anti-CD3-spot (**e**) surfaces. Above in each panel, representative images of CD11a-YFP with fluorescent SA spots either shown or removed to reveal the CD11a-YFP pattern (scale bars = 7 μm). Below, fluorescence intensity of SA (left) and CD11a-YFP (right) computed by averaging z-projection images centered on the SA spot (see "Methods"). $N = 36$ for (**d**) and $N = 30$ for (**e**). Scale bars = 2 μm. All data are representative of at least two independent experiments. Source data are provided as a Source Data file.

in zones where pMHC and ICAM-1 were either copresented (Dual-spot) or closely apposed (ICAM-spot and Antigen-spot) further supports a critical role for the coengagement of the TCR and LFA-1 in guiding cytotoxic secretion and suggests that permissive secretory domains of receptor coengagement can be substantially smaller than the IS itself.

Although all three micropatterned substrates elicited targeted degranulation, responses to the Dual-spot and Antigen-spot configurations were markedly more focused than what we observed on ICAM-spot surfaces (Fig. 4c). To explore the basis for this difference, we imaged CTLs expressing a fluorescent form of the LFA-1 α-chain (CD11a-YFP) on surfaces containing focal LFA-1 or TCR ligands. Because the CD11a-YFP CTLs were derived from polyclonal animals, we used anti-CD3 antibodies to engage the TCR. On ICAM-spot surfaces (anti-CD3 in the background), LFA-1 tended to localize to the micropatterned ICAM-1 (Fig. 4d), consistent with ligand recognition. The Anti-CD3-spot configuration, however, induced even stronger focal accumulation of LFA-1 (Fig. 4e), which was surprising considering that ICAM-1 was coated in the background of these surfaces. Interestingly, LFA-1 recruitment on Anti-CD3-spot substrates was most apparent not over the anti-CD3 spot itself but in the surrounding 3–4 μm neighborhood (Fig. 4e). This could potentially reflect inside-out integrin activation and clustering induced by local TCR signaling[27]. The unexpectedly robust accumulation of LFA-1 to areas of focal TCR stimulation potentially explains

why Antigen-spot surfaces elicited more focused degranulation than their ICAM-spot counterparts.

**Degranulation occurs in regions of LFA-1-dependent force exertion.** Within the IS, both the TCR and LFA-1 are subjected to F-actin-dependent pulling forces, which are thought to drive the formation of catch bonds between each receptor and its respective ligand, promote conformational changes, and induce signal transduction[18,19,22,23]. Given the importance of these forces for the function of each receptor, we reasoned that they might also play a role in guiding cytotoxic secretion. To investigate this hypothesis, we employed Förster resonance energy transfer (FRET)-based molecular tension probes (MTPs)[43] specific for the TCR and LFA-1. Each MTP comprised a stimulatory ligand (pMHC or ICAM-1) attached to a DNA hairpin containing a fluorophore at one end (Atto647N or Cy3B, respectively) and a quencher (BHQ-2) at the other (Fig. 5a). When folded at a resting state, MTPs do not fluoresce due to the close proximity between quencher and fluorophore. Applied forces capable of unwinding the hairpin (in this case, 4.7 pN) pull the quencher and fluorophore apart, dramatically increasing fluorescence. Consistent with prior reports[44,45], surfaces coated with pMHC-MTPs and ICAM-1-MTPs induced IS formation by OT-1 CTLs and the exertion of dynamic forces through both LFA-1 and the TCR (Fig. 5b and Supplementary Movie 4), which we visualized by

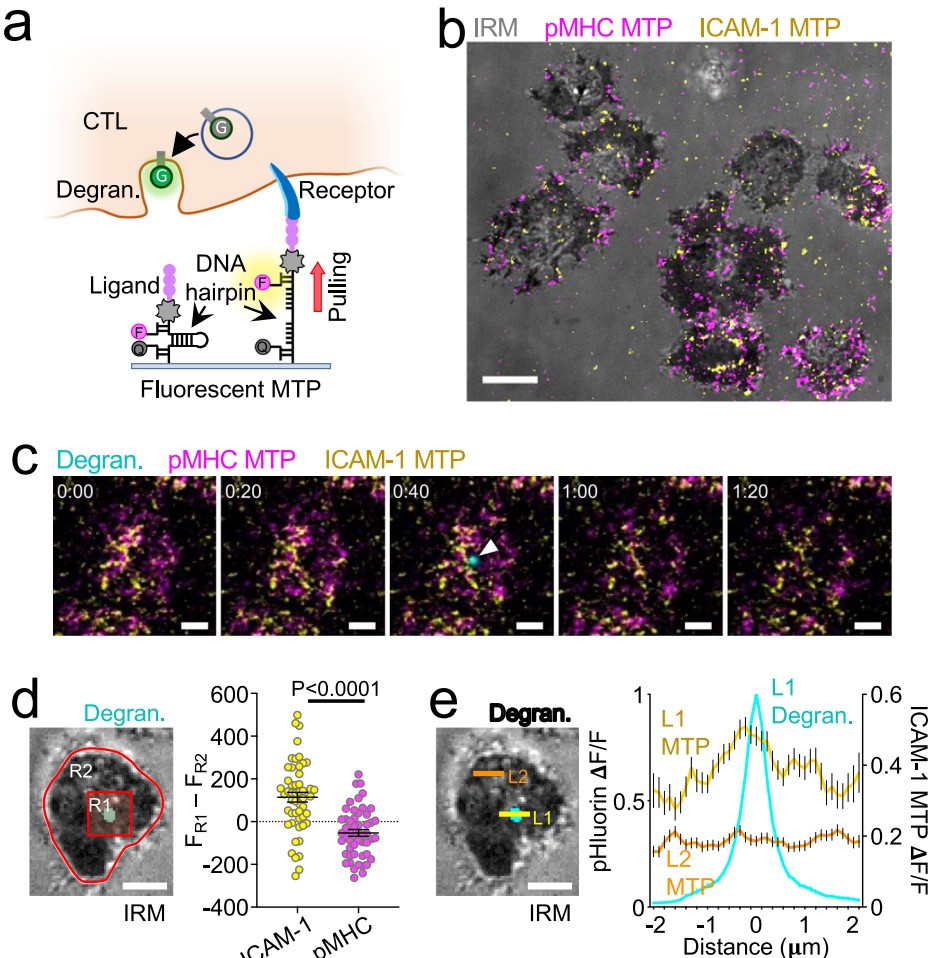

**Fig. 5 LFA-1 pulling forces define degranulation domains. a** Measuring correlations between degranulation (Degran.) and receptor-specific pulling forces with MTPs. F fluorophore, Q quencher, G pHluorin. **b–e** OT-1 CTLs expressing pHluorin-Lamp1 were imaged by TIRF microscopy on glass surfaces coated with pMHC (H-2K$^b$-OVA) and ICAM-1 MTPs. **b** Representative image of pMHC-MTP and ICAM-1-MTP signals, overlaid onto the corresponding IRM image. Scale bar = 5 μm. **c** Time-lapse montage showing a representative degranulation event (indicated by a white arrowhead) together with pMHC-MTP and ICAM-1-MTP signals. Time in M:SS is shown at the top left corner of each image. Scale bars = 2 μm. **d** Left, image of a representative degranulation event, overlaid onto the corresponding IRM image. Regions defining the degranulation subdomain (R1) and the entire IS (R2) are indicated. Scale bar = 2 μm. Right, differences in mean fluorescence intensity between R1 and R2 at the moment of degranulation are shown for the indicated MTPs. N = 52 for each sample. Error bars signify SEM. P value calculated by unpaired, two-tailed Student's t test. **e** Left, the image of a representative degranulation event, overlaid onto the corresponding IRM image. Linescans sampling the degranulating (L1) and inactive (L2) domains are indicated. Scale bar = 2 μm. Right, normalized ICAM-1-MTP fluorescence along L1 and L2 at the moment of degranulation (see "Methods"). The pHluorin-Lamp1 (Degran.) signal along L1 is shown for reference. Error bars signify SEM. N = 122 cells, pooled from two independent experiments. All other data are representative of at least two independent experiments. Source data are provided as a Source Data file.

time-lapse imaging. To measure the association between degranulation and receptor-specific forces, we used CTLs expressing pHluorin-Lamp1 to record exocytic events elicited by stimulatory MTPs (Fig. 5c). The mean MTP fluorescence in the immediate vicinity of each event (2-μm box) was then compared with the mean fluorescence of the entire IS. This approach revealed a marked enrichment of ICAM-1-MTP signal in the degranulation zone (Fig. 5d), indicative of a spatial correlation between cytotoxic secretion and force exertion through LFA-1. pMHC-MTP pulling was not associated with degranulation in this way (Fig. 5d), arguing against a role for the TCR as a critical force-bearing receptor in this context. To further characterize the pattern of LFA-1 mechanics, we examined ICAM-1-MTP fluorescence along linescans bisecting the degranulation peak. Mean LFA-1 forces reached a local maximum in the 1 μm diameter region surrounding each event (Fig. 5e), consistent with the idea that mechanically active LFA-1 defines permissive zones for

cytotoxic secretion. A degranulation zone of this size would accommodate the approach of a typical lytic granule (0.5–1 μm in diameter)[46].

**Integrin-dependent forces promote synaptic degranulation and killing.** Integrins are coupled to the cytoskeleton via talin, a mechanosensitive scaffolding protein that binds both the β-subunit tail and F-actin[28] (Fig. 6a). To evaluate the importance of the integrin–cytoskeletal linkage for synaptic force exertion, we used CRISPR/Cas9 to deplete talin from OT-1 CTLs (Supplementary Figs. 7a, 12) and then compared the physical output of these cells to that of controls expressing a nontargeting guide RNA. Although talin-deficient CTLs expressed normal levels of cell-surface LFA-1 (Supplementary Figs. 1e, 7b), their capacity to exert pulling forces against ICAM-1 MTPs was dramatically reduced (Fig. 6b, c and Supplementary Movies 5 and 6). By

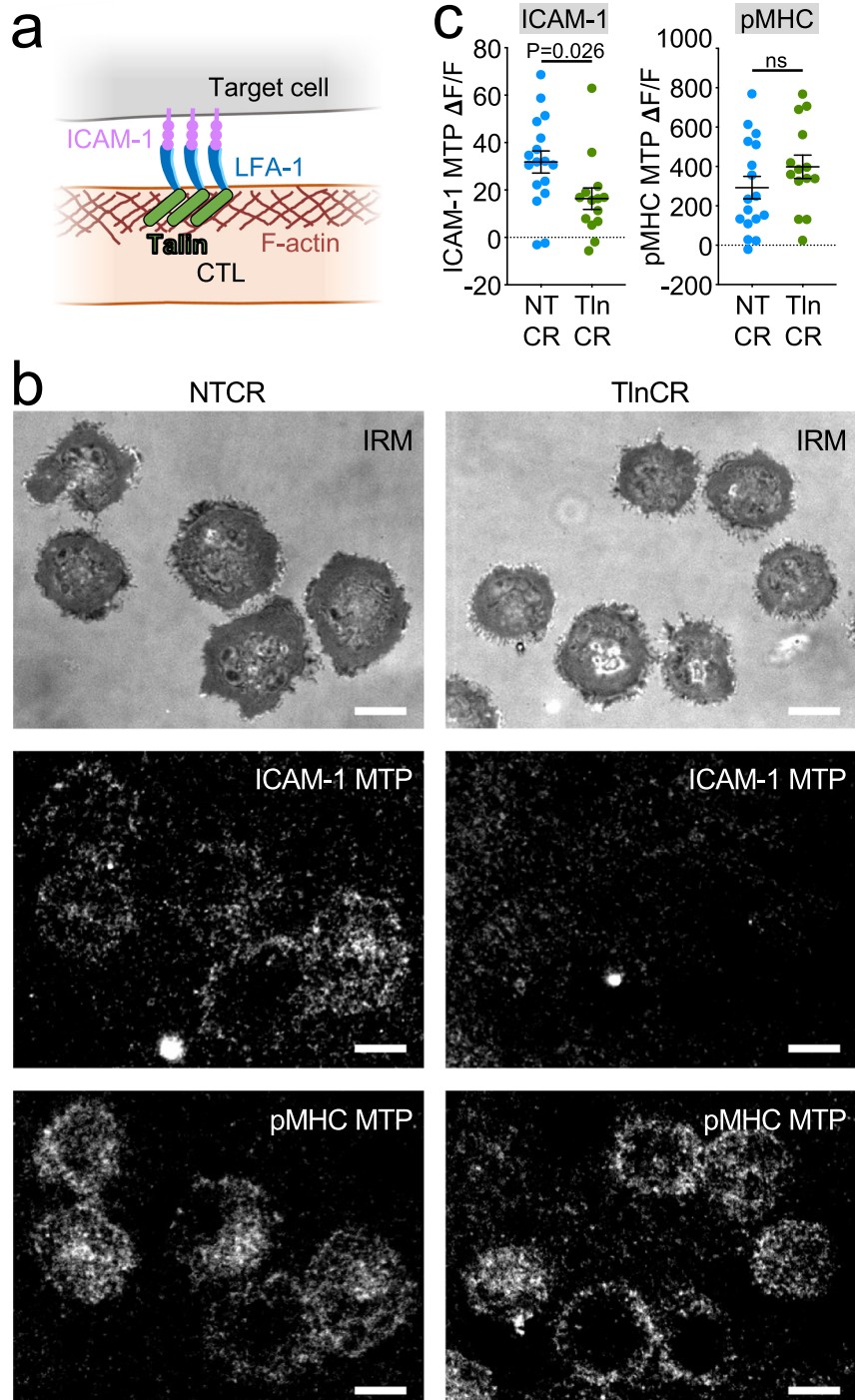

**Fig. 6 Talin is required for LFA-1-mediated force exertion. a** Talin couples ligand-bound integrins to the F-actin cytoskeleton. **b, c** OT-1 Cas9 CTLs expressing talin gRNA (TlnCR) or control nontargeting gRNA (NTCR) were imaged on surfaces bearing the indicated stimulatory MTPs. **b** Representative images showing ligand-specific pulling in synapses. Scale bars = 5 μm. **c** ICAM-1-MTP (left) and pMHC (H-2K$^b$-OVA)-MTP (right) signals, assessed 30 min after the addition of CTLs to the MTP surface. $N \geq 14$ cells for each sample. Error bars signify SEM. $P$ values calculated by unpaired, two-tailed Student's $t$ test (ns not significant). All data are representative of at least two independent experiments. Source data are provided as a Source Data file.

contrast, pMHC-MTP pulling forces were unchanged (Fig. 6b, c and Supplementary Movies 5 and 6), indicating that the mechanical contribution of talin is restricted to LFA-1 in this system. In cocultures with OVA-loaded RMA-s cells, CTLs lacking talin exhibited sharply reduced degranulation and target cell lysis (Fig. 7a, b and Supplementary Fig. 8a, b), implying a central role for integrin adhesions in both processes. These loss-of-function phenotypes were not rescued by PMA/Iono,

indicating that they were not caused by impaired T-cell activation. Consistent with this interpretation, depletion of talin did not affect TCR-induced MAPK and PI3K signaling (Supplementary Figs. 7c, 12), and it had only modest effects on CD69 responses (Fig. 7c and Supplementary Fig. 8c). TCR-induced F-actin accumulation and centrosome polarization to the IS were also normal (Supplementary Fig. 9). The disproportionately large effect of talin deficiency on cytotoxic secretion was particularly

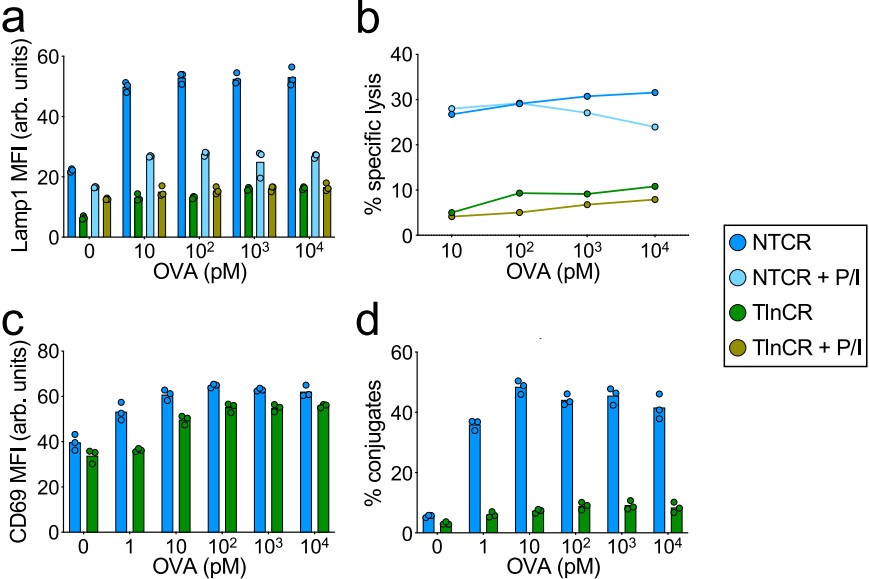

**Fig. 7 Talin is required for CTL degranulation and cytotoxicity. a–d** OVA-loaded RMA-s target cells were mixed with OT-1 Cas9 CTLs expressing talin gRNA (TlnCR) or control nontargeting gRNA (NTCR). PMA/Iono (P/I) was applied to some samples in order to drive TCR-independent CTL activation. **a** Lamp1 exposure (degranulation), measured 90 min after CTL-target cell mixing (N = 4 replicate experiments). **b** Target cell killing, measured 4 h after CTL-target cell mixing (N = 2 replicate experiments). **c** CD69 expression, measured 90 min after CTL-target cell mixing (N = 2 replicate experiments). **d** Conjugate formation, measured 90 min after CTL-target cell mixing (N = 2 replicate experiments). Data points in (**a, c, d**) represent technical triplicate measurements from an individual experiment. Data points in (**b**) represent mean values calculated from technical triplicates. Source data are provided as a Source Data file. Replicate studies are shown in Supplementary Fig. 8.

obvious in two-dimensional plots of Lamp1 and CD69 (Supplementary Fig. 8e), which confirmed that, for a given level of activation, CTLs lacking talin consistently degranulated more weakly than nontargeting controls. Talin depletion also suppressed CTL-target cell conjugate formation (Fig. 7d and Supplementary Fig. 8d), similar to the effects of LFA-1 blockade.

Collectively, these results suggested an important role for talin in LFA-1-dependent IS mechanics and cytotoxicity. They fell short, however, of definitively showing that force exertion through LFA-1 is critical for synaptic degranulation. This is because talin not only couples integrins to the cytoskeleton (outside-in signaling) but also induces integrin extension into an intermediate affinity state (inside-out signaling) (Fig. 8a). Hence, the talin loss-of-function phenotypes described above could simply have resulted from an inability of LFA-1 to bind ICAM-1, rather than a defect in mechanotransduction. Indeed, CTLs lacking talin exhibited only weak adhesion to ICAM-1-coated surfaces (Supplementary Fig. 10a), and their capacity to migrate along these surfaces was similarly impaired (Supplementary Fig. 10b).

To interrogate talin-dependent force exertion specifically, an experimental strategy was required that could separate the cytoskeletal coupling function of talin from its capacity to drive integrin extension. Prior studies had demonstrated that the N-terminal head domain of talin mediates integrin binding and integrin activation, while the C-terminal rod domain associates with F-actin[28]. Accordingly, we retrovirally transduced CTLs lacking endogenous talin with a construct encoding the talin head, generating cells that effectively lacked only the talin rod (Fig. 8a and Supplementary Figs. 7a and 12). This strategy partially restored both adhesion to and migration on top of ICAM-1-coated surfaces (Supplementary Fig. 10a, b), indicative of rescued inside-out signaling. To further validate this approach, we examined IS formation on supported lipid bilayers containing laterally mobile pMHC and ICAM-1. CTLs lacking talin spread

poorly on these surfaces, but overexpression of the talin head largely reversed this phenotype (Fig. 8b, c). The capacity of talin perturbations to modulate cell spreading depended specifically on ICAM-1 binding, as neither talin deletion nor talin head expression altered IS size on bilayers coated with pMHC alone (Supplementary Fig. 10c). Collectively, these data indicated that the talin head domain could indeed restore LFA-1 activation in talin-deficient cells, which was in line with prior work[47]. Next, we applied MTP imaging to assess the effects of this rescue approach on receptor-specific force exertion. Talin head expression enhanced LFA-1 dependent pulling forces in control CTLs (Fig. 8d, e), possibly reflecting increased integrin extension. The same strategy, however, completely failed to restore the mechanical activity of talin-deficient CTLs (Fig. 8d, e), confirming the importance of talin-F-actin interactions for force exertion through integrins. Talin head expression was similarly ineffective at rescuing talin-dependent cytotoxic function. CTLs lacking just the talin rod domain (talin CRISPR plus talin head) exhibited degranulation and conjugate defects that were indistinguishable from those of CTLs lacking the entire protein (Fig. 8f, g and Supplementary Fig. 10d). Taken together, these results indicate that outside-in mechanotransduction from LFA-1 to the cortical F-actin cytoskeleton is critical for IS mechanics and, furthermore, that force exertion through LFA-1, rather than the TCR, imposes spatiotemporal control over CTL degranulation.

Not all cells express LFA-1 ligands, raising the question of whether integrin mechanotransduction controls CTL-mediated killing across a broad spectrum of targets. The capacity of talin deficiency to interrogate integrin function independently of LFA-1 allowed us to address this issue. B16F10 melanoma cells do not express ICAM-1, implying that they cannot engage LFA-1 across the IS (Fig. 9a, b). Nevertheless, they are reasonable targets for OT-1 CTLs, eliciting robust degranulation and cytotoxicity responses in the presence of OVA. LFA-1 blockade failed to inhibit either of these responses (Fig. 9c and Supplementary

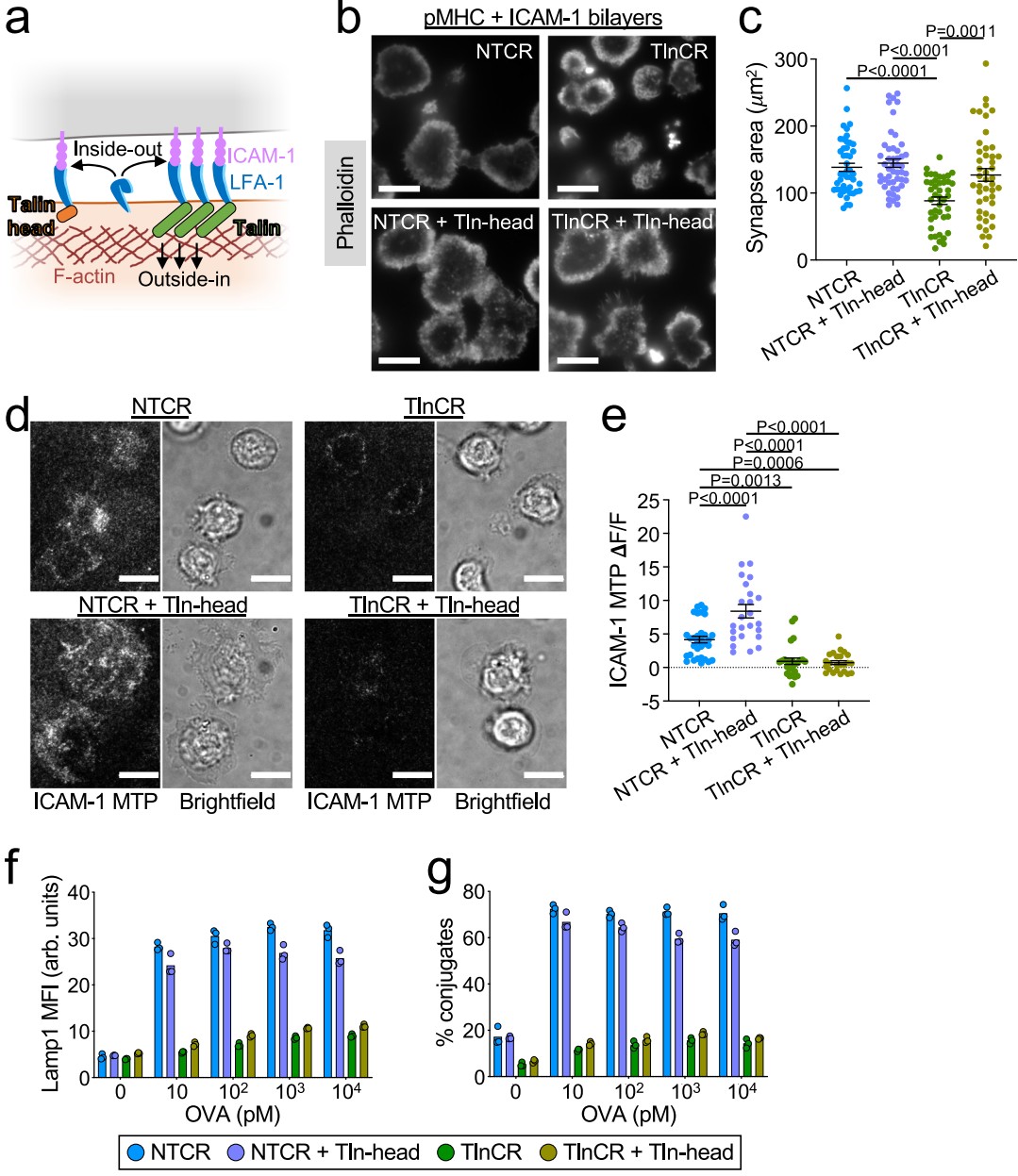

**Fig. 8 Cytoskeletal coupling is critical for the mechanical and cytotoxic function of talin. a** The talin-head induces inside-out integrin activation without coupling integrins to the cytoskeleton (outside-in). **b, c** OT-1 Cas9 CTLs expressing talin gRNA (TlnCR) or control nontargeting gRNA (NTCR) ± the talin head domain (Tln head) were fixed on pMHC + ICAM-1-coated bilayers and stained for phalloidin. **b** Representative TIRF images of synaptic F-actin. **c** Quantification of IS area ($N \geq 43$ cells for each sample), representative of two independent experiments. **d, e** OT-1 CTLs expressing the indicated gRNAs ± Tln-head were imaged on surfaces containing pMHC and ICAM-1 MTPs. **d** Representative brightfield and ICAM-1 MTP images. **e** Quantification of ICAM-1 MTP pulling forces, assessed 30 min after the addition of CTLs to the MTP surface. $N \geq 24$ cells for each sample, representative of two independent experiments. **f, g** OVA-loaded RMA-s target cells were mixed with OT-1 Cas9 CTLs expressing the indicated gRNAs ± Tln-head. **f** Lamp1 exposure (degranulation), measured 90 min after CTL-target cell mixing ($N = 2$ replicate experiments). **g** Conjugate formation, measured 90 min after CTL-target cell mixing ($N = 2$ replicate experiments). Data in (**f, g**) were derived from technical triplicates. All scale bars = 10 μm, and all error bars signify SEM. *P* values in (**c, e**) calculated by one-way ANOVA with Tukey correction. Data points (**f, g**) represent technical triplicate measurements from an individual experiment. Source data are provided as a Source Data file. Replicate studies for (**f, g**) are shown in Supplementary Fig. 10.

Fig. 11a), consistent with the idea that LFA-1 is not involved in the recognition and killing of B16F10 cells. By contrast, talin depletion abrogated both target cell lysis and degranulation (Fig. 9d and Supplementary Fig. 11b), strongly suggesting that integrins other than LFA-1 contribute to the killing of ICAM deficient targets. We conclude that integrin-mediated control of cytotoxic secretion is likely to be a general feature of the IS.

## Discussion

Taken together, our data indicate that degranulation occurs at permissive secretory sites within the IS that are defined by mechanically active integrins (Fig. 10). TCR signaling plays a critical role in this process by inducing close contact formation and also by triggering Ca²⁺ influx, which is required for granule fusion[48–50]. In the absence of integrin-dependent force exertion,

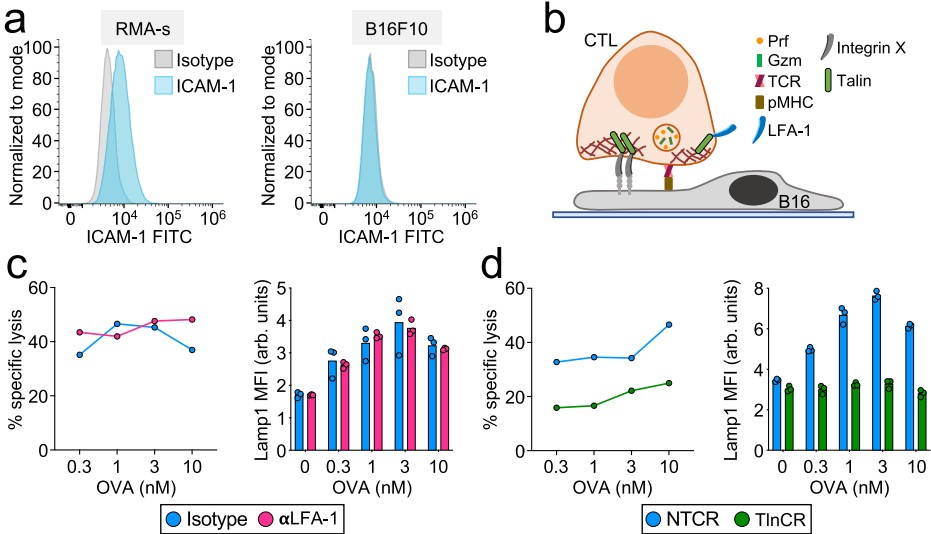

**Fig. 9 Talin, but not LFA-1, is required for CTL-mediated killing of B16F10 cells. a** Representative histograms showing ICAM-1 expression in RMA-s (left) and B16F10 (right) cells ($N = 2$ replicate experiments). **b** Model of CTL-mediated killing of B16F10 cells that requires both the TCR and an unidentified integrin X. **c** OVA-loaded B16F10 target cells were mixed with OT-1 CTLs in the presence of LFA-1-blocking antibody (αLFA-1) or isotype control. Left, specific lysis, measured after 4 h ($N = 2$ replicate experiments). Right, Lamp1 exposure, measured after 90 min ($N = 2$ replicate experiments). **d** OVA-loaded B16F10 target cells were mixed with OT-1 Cas9 CTLs transduced with talin gRNA (TlnCR) or control nontargeting gRNA (NTCR). Left, specific lysis, measured after 4 h ($N = 2$ replicate experiments). Right, Lamp1 exposure, measured after 90 min ($N = 2$ replicate experiments). Data points in degranulation graphs represent technical triplicate measurements from an individual experiment. Data points in killing assays represent mean values calculated from technical triplicates. Source data are provided as a Source Data file. Replicate studies for (**c**, **d**) are shown in Supplementary Fig. 11.

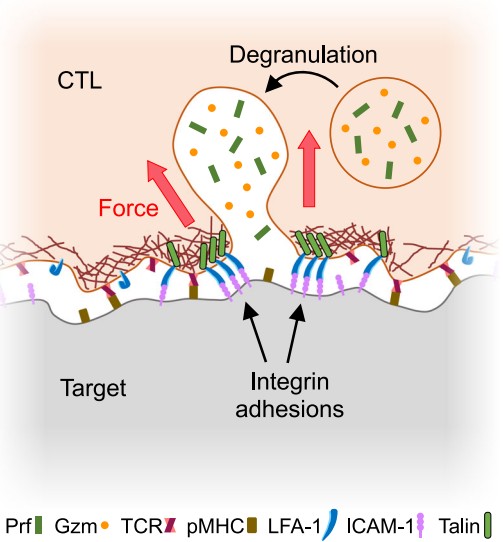

**Fig. 10 Mechanical licensing of degranulation by integrin adhesions.** Lytic granules fuse within synaptic subdomains defined by ligand-bound integrins under tension.

however, TCR signaling alone is insufficient for robust cytotoxic secretion. Indeed, our results imply that one of the major ways that the TCR promotes killing is by local activation of LFA-1.

Canonically, degranulation is thought to occur at the center of the IS, close to where ligand-bound activating receptors, like the TCR, are internalized for recycling[51–53]. This is an attractive model because the central domain, also called the central supramolecular activation cluster (cSMAC), contains relatively low levels of cortical F-actin, which in principle makes its plasma membrane more accessible to approaching granules. Work from multiple labs, however, indicates that this canonical model oversimplifies the secretory process. High-resolution imaging

studies have shown that the cSMAC contains substantial amounts of residual F-actin and that degranulation occurs in small, transient openings in the F-actin cortex[42,54]. These openings often appear just adjacent to F-actin-rich, mechanically active protrusive structures[20], implying that there is a functional role for F-actin in granule fusion beyond simply getting out of the way. We now demonstrate that integrins, which couple directly to the F-actin cortex, mechanically license membrane subdomains for degranulation. In doing so, we confirm and extend prior studies documenting lytic granule accumulation and perforin release in synaptic regions defined by integrin engagement[38,39,55]. Hence, multiple strands of investigation now point toward a revised model in which dynamic interplay between integrins and F-actin remodeling guides degranulation within the IS.

Within synaptic secretory zones, the integrin-mediated mechanical coupling between the CTL and target cell would enable the CTL to "feel" the presence of the target by pushing or pulling against it. Selective degranulation at sites of synaptic force exertion would thereby ensure that perforin and granzyme are released only in areas of close CTL-target cell apposition, minimizing risk to innocent bystander cells. Synaptic forces have also been shown to potentiate the pore-forming activity of perforin, likely by straining the target cell membrane[16,20]. Guiding granules to mechanically active zones within the IS would be expected to facilitate this process by spatiotemporally coupling the delivery of perforin with synaptic force. Hence, mechanical licensing of cytotoxic secretion by integrins is expected to promote both the specificity and the potency of killing responses.

In the absence of LFA-1 ligands, we speculate that other integrins might assume the licensing role. Indeed, the fact that B16F10 cell killing requires talin, but not LFA-1, strongly implies the existence of alternative integrin activators. CTLs express both VLA-4 ($\alpha_4\beta_1$) and CD103 ($\alpha_E\beta_7$), which recognize protein ligands (VCAM-1/2 and E-Cadherin, respectively) found on subsets of potential target cells. CD103 is a particularly interesting candidate, as it has been shown to promote granule polarization toward

E-Cadherin expressing tumor cells[56]. There are also indications that talin can function independently of integrins[57], raising the possibility that other classes of adhesion receptor might act through talin to guide degranulation against specific classes of target.

Although CTLs expressing the talin head domain alone spread normally on stimulatory bilayers, full-length talin was essential for conjugate formation with bona fide target cells. Taken together, these observations suggest that inside-out integrin activation is sufficient for engaging ligands that can diffuse freely over the target surface, but that mechanical coupling between integrins and the cytoskeleton is required for a strong association with ligands that pull back against applied forces due to surface rigidity, constrained lateral mobility, or the presence of a glycocalyx. This conclusion is in line with prior work in fibroblasts showing that the talin head domain can drive transient cell spreading on fibronectin, but that the full-length protein is necessary for long-lived substrate adhesion and normal cell morphology[47]. In addition to enabling mechanotransduction, linking transmembrane proteins to the F-actin cortex also allows cells to coordinate receptor movement and to modulate the cooperativity of ligand binding[58]. F-actin thus serves as a critical regulator and integrator of adhesion and communication across cell–cell interfaces.

That talin-deficient CTLs fail to degranulate despite exhibiting normal centrosome polarization is consistent with prior work showing that the centrosome is dispensable for synaptic secretion[13,14]. Our integrin-based licensing model explains these observations by providing a mechanism for imposing directionality that does not depend on the microtubule cytoskeleton. That being said, our results do not exclude an important role for the centrosome and microtubules in enhancing the speed and efficiency of cytotoxic secretion. Indeed, we have shown that microtubule depletion, while failing to disrupt the directionality of degranulation, nevertheless profoundly reduces the magnitude of the secretory response[14]. Accordingly, we favor a model in which the centrosome and its associated microtubules transport granules into the IS neighborhood, at which point their site of fusion is determined by mechanically active integrins. In adherent cells, a subset of microtubule tips anchor close to integrin adhesions[59], and disruption of these adhesions has been found to induce dramatic remodeling of the microtubule cytoskeleton[47]. Hence, it is tempting to speculate that microtubule tips might directly target zones of integrin engagement within the IS, thereby enabling the delivery of lytic granules to fusion competent membrane domains.

Precisely how the granule trafficking machinery identifies zones of integrin engagement remains to be seen, but presumably involves the conversion of integrin mechanics into a molecular identifier of some sort. An obvious candidate for the generation of such an identifier is the adhesion complex itself, which not only couples integrins to the cytoskeleton but also propagates a variety of outside-in signals. In fact, prior work has specifically implicated outside-in signaling in target cell killing by NK cells[60]. Integrin adhesions activate a variety of enzymatic effectors, including lipid-modifying enzymes like PI3K and phospholipase c-γ[32,61,62]. One or more of these enzymes could alter local membrane composition so as to drive the docking and/or priming of lytic granules. Lipid control mechanisms are a recurring theme in regulated secretion[63], and therefore seem likely to modulate lymphocyte degranulation.

Synaptic mechanotransduction requires not only that cytoskeletal forces be applied through mechanosensitive cell-surface receptors, but also that the opposing ligand-presenting surface be rigid enough to resist these pulling forces. Accordingly, stiffer ligand-presenting substrates or cells have been found to elicit stronger lymphocyte activation in vitro than softer surfaces coated with the same ligands[64–67]. We have recently found that this paradigm governs anti-tumor responses in vivo, with cytotoxic lymphocytes preferentially targeting stiffer cancer cells in the metastatic niche[68]. This mechanical form of immunosurveillance, which we have termed mechanosurveillance, also appears to operate during checkpoint blockade immunotherapy. The mechanically licensed degranulation model we have developed here provides a mechanistic basis for mechanosurveillance and highlights a potentially critical role for integrins in the process.

The coupling of mechanical input to secretory output is unlikely to be unique to cytotoxic lymphocytes. Indeed, one can imagine analogous mechanisms regulating other mechanically active processes, like phagocytosis and cell–cell fusion, that involve secretion and/or polarized membrane remodeling. We anticipate that biophysical analysis of systems like these will further illuminate the scope and functional relevance of mechano-secretory crosstalk in communicative cell–cell interactions.

## Methods

**Constructs**. The retroviral expression construct for pHluorin-Lamp1 has been described[42]. A CRISPR gRNA construct targeting talin was prepared according to a published protocol[69] using the following targeting sequence: 5′-GCTTGGCTTG TGAGGCCAGT-3′. A nontargeting control construct was also prepared using the sequence 5′-GCGAGGTATTCGGCTCCGCG-3′. After PCR amplification, DNA fragments encoding these guide sequences were subcloned into the pMRIG vector using the BamHI and MfeI restriction sites. cDNA encoding the talin head domain (a.a. 1–433) fused at its 5′ end to GFP was amplified from plasmid DNA (Addgene 32856) and then ligated into a retroviral expression vector (p2MSCV) using the NotI and BamHI sites.

**Proteins**. Class I MHC proteins (H-2K$^b$ and H-2D$^b$) were overexpressed in E. Coli, purified as inclusion bodies, and refolded by rapid dilution in the presence of β2-microglobulin and either OVA (for H-2K$^b$) or KAVYDFATL (KAVY, for H-2D$^b$). Monomeric MHC proteins were biotinylated using the BirA enzyme and purified by size exclusion chromatography. The extracellular domain of mouse ICAM-1 (a.a. 28–485, polyhistidine-tagged) was expressed by baculoviral infection of Hi-5 cells and purified by Ni$^{2+}$ chromatography. After BirA-mediated biotinylation, the ICAM-1 was further purified by size exclusion.

**Animals and cells**. The animal protocols used for this study were approved by the Institutional Animal Care and Use Committee of Memorial Sloan Kettering Cancer Center. Animals were maintained under specific pathogen-free conditions. Different strains were housed separately prior to euthanasia by carbon dioxide asphyxiation and subsequent lymphocyte extraction. Typically, one mouse was sufficient to generate lymphocytes for a week's worth of experiments. Both male and female mice were used. Primary OT-1 CTL blasts were prepared by pulsing splenocytes from an OT-1 αβTCR transgenic animal (Jackson Labs #003831) with 100 nM OVA in RPMI medium containing 10% (vol/vol) FCS. Cells were supplemented with 30 IU/mL IL-2 after 24 h and were split as needed in RPMI medium containing 10% (vol/vol) FCS and IL-2. To generate OT-1 blasts with talin modifications, splenocytes from Cas9 knockin mice[70] (Jackson Labs #026179) crossed onto the OT-1 background were stimulated with OVA as described above and then transduced 24 h later with retrovirus encoding gRNA against talin, nontargeting control gRNA, or the talin head domain. T cell blasts were prepared from polyclonal cd11a$^{-/-}$ mice (Jackson Labs #042053), cd11a-yfp knockin mice[71], and C57BL/6 controls (Jackson Labs #000664) by culturing splenocytes on plate-bound anti-CD3 (clone 2C11) and anti-CD28 (clone 37.51). Cells were supplemented with 30 IU/mL IL-2 after 24 h and were split as needed in RPMI medium containing 10% (vol/vol) FCS and IL-2 for seven days. This protocol typically generates a ~4:1 mixture of CD8$^+$ versus CD4$^+$ effectors. RMA-s cells and the human CTL clone OM265 CMV were maintained in RPMI containing 10% (vol/vol) FCS. B16F10 cells were maintained in DMEM medium containing 10% (vol/vol) FCS.

**Retroviral transduction**. Phoenix E cells were transfected with expression vectors and packaging plasmids using the calcium phosphate method. Ecotropic viral supernatants were collected after 48 h at 37 °C and added to 1.5 × 10$^6$ OT-1 blasts 24 h after primary peptide stimulation. Mixtures were centrifuged at 1400 × g in the presence of polybrene (4 μg/mL) at 35 °C, after which the cells were split 1:3 in RPMI medium containing 10% (vol/vol) FCS and 30 IU/mL IL-2 and allowed to grow for an additional 4–6 days.

**Traction force microscopy**. Arrays of PDMS (Sylgard 184; Dow Corning) micropillars (0.7 μm in diameter, 6 μm in height, spaced hexagonally with a 2-μm center-to-center distance) were cast onto glass coverslips using the inverse PDMS mold method[72]. After an ethanol wash and stepwise exchange into phosphate-buffered saline (PBS), pillars were stained with fluorescently labeled streptavidin (20 μg/mL Alexa Fluor 647, ThermoFisher Scientific) for 2 h at room temperature. Following additional PBS washes, the arrays were incubated with biotinylated H-2K$^b$-OVA ± ICAM-1 (10 μg/mL each) overnight at 4 °C. The pillars were then washed into RPMI containing 5% (v/v) FCS and lacking phenol red for imaging. T cells stained with Alexa Fluor 488-labeled anti-CD45.2 Fab (clone 104-2) were then added to the arrays and imaged using an inverted fluorescence microscope (Olympus IX-81) fitted with a 100× objective lens and a mercury lamp for excitation. Images in the 488 nm (CTLs) and 647 nm (pillars) channels were collected every 15 s using MetaMorph software (version 7.8.2.0).

**Antibody blockade and pharmacological activation/inhibition**. To assess the importance of LFA-1, CTLs were pre-incubated with LFA-1 blocking antibody (20 μg/mL, Clone M17/4, BioXCell) or an IgG2aκ isotype control antibody (20 μg/mL, Clone RTK2758, BioLegend) at 37 °C for 5 min before the addition of target cells/stimulatory beads. The final concentration of antibodies during the assay was 10 μg/mL. Human LFA-1 was blocked using the anti-CD18 clone TS1/18 (20 μg/mL, Invitrogen), with a mouse IgG2a antibody (Sigma) serving as an isotype control. To induce T-cell activation independently of the TCR, CTLs were pre-incubated with phorbol myristate acetate (PMA, 20 ng/mL, Sigma Aldrich) and the Ca$^{2+}$ ionophore A23187 (2 μM, Tocris Bioscience) at 37 °C for 5 min before the addition of target cells/stimulatory beads, yielding final concentrations of 10 ng/mL PMA and 1 μM A23187. Additional information about antibodies may be found in Supplementary Table 1.

**Functional assays**. To measure cytotoxicity, RMA-s target cells were labeled with CellTrace Violet (CTV), loaded with increasing concentrations of OVA, and mixed 3:1 with PKH26-stained OT-1 CTLs in a 96-well V-bottomed plate. Specific lysis of CTV$^+$ target cells was determined by flow cytometry after 4 h at 37 °C[73]. To quantify degranulation, OT-1 CTLs were mixed with RMA-s target cells as described above and incubated at 37 °C for 90 min in the presence of eFluor 660-labeled anti-Lamp1 (clone eBio1D4B; eBioscience). Cells were then stained with FITC-labeled anti-CD69 (clone H1.2F3) and subjected to flow cytometric analysis to quantify Lamp1 and CD69 staining. To measure conjugate formation, labeled OT-1 CTLs and RMA-s targets were mixed 1:1, lightly centrifuged (100 × g) to encourage cell contact, and incubated 90 min at 37 °C. Cells were then resuspended in the presence of 2% paraformaldehyde, washed in fluorescence-activated cell sorting buffer (PBS + 4% FCS), and analyzed by flow cytometry. Conjugate formation was quantified as (PKH26$^+$CTV$^+$)/(PKH26$^+$). For B16F10 killing assays, B16F10 targets were cultured overnight on fibronectin and then pulsed with varying concentrations of OVA for 2 h. OT-1 CTLs were added at an 8:1 E:T ratio and incubated for 3–4 h at 37 °C in RPMI medium supplemented with 10% FCS and IL-2 (30 IU/mL). Target cell death was quantified with an LDH (lactate dehydrogenase) cytotoxicity assay kit (Clontech) using the manufacturer's recommended protocol. To measure intracellular granzyme B depletion, OT-1 CTLs were mixed 1:3 with OVA-loaded RMA-s cells and incubated for 4–6 h at 37 °C. Intracellular granzyme B levels were then measured by flow cytometry after fixation, permeabilization, and staining with Alexa 647-labeled anti-granzyme B (clone GRB11, BioLegend). All functional assays were performed in triplicate. To quantify ICAM-1 expression, RMA-s or B16F10 cells were stained with a fluorescently labeled anti-ICAM-1 antibody CD54 (Clone YN1/1.7.4, BioLegend) or isotype control antibody (Clone RTK4530, BioLegend). LFA-1 expression was measured using an antibody against CD11a (Clone 17/4, BioLegend). To quantify antigen-induced degranulation by human CTLs, purified HLA-A2$^+$ CD4$^+$ T-cell targets were loaded with a range of NV9 (NLVPMVATV) peptide doses and then mixed with OM265 CMV T cells at an E:T ratio of 1:2 in the presence of PE-Cy7 labeled anti-Lamp1 antibody (Clone H4A3). After 4 h, the samples were stained with anti-CD4-BV786 (Clone RPA-T4), anti-CD8-BV605 (Clone RPA-T8), and Live/Dead Aqua (Invitrogen), followed by flow cytometric analysis. To assess proliferation, day 7 OT-1 CTLs were stained with CellTrace Violet at room temperature for 20 min, washed in serum-containing medium, and then incubated with irradiated OVA-loaded C57BL/6 splenocytes (0.5 × 10⁶ CTL with 4.0 × 10⁶ splenocytes) in the presence of 10 μg/mL anti-LFA-1 or an isotype control antibody. Subsequent dilution of CellTrace Violet was assessed by flow cytometry. Additional information about antibodies may be found in Supplementary Table 1.

**CTL activation with stimulatory beads**. Streptavidin-conjugated polystyrene beads (Spherotech) were coated with 1 μg/mL biotinylated ICAM-1 and/or various concentrations of biotinylated H-2K$^b$-OVA. Nonstimulatory pMHC (H-2D$^b$) was used if necessary to adjust the total biotinylated protein concentration of each mixture to 2 μg/mL. After overnight incubation at 4 °C, the excess unbound protein was washed out and the beads were transferred into RPMI medium containing 10% (vol/vol) FCS (± phenol red) for use in experiments. For immunoblot analysis of signaling, beads were mixed with OT-1 CTLs at a 1:1 ratio. For functional studies (e.g., degranulation), the CTL to bead ratio was 1:3. For 2 bead stimulation (e.g.,

Fig. 3), Nile Red and Purple streptavidin beads were coated as described above with H-2K$^b$-OVA and ICAM-1 in either *cis* or *trans* configurations. H-2D$^b$ was used to fill empty spaces in the *trans* configuration and also to generate "dummy" coated beads. In each experimental condition, CTLs were mixed with two kinds of beads at a 1:3:3 ratio. Degranulation and CD69 upregulation were quantified by flow cytometry as described above.

**Micropatterning experiments**. PDMS stamps for imprinting 2 μm diameter spots in 10 μm center-to-center square arrays were prepared using microfabricated silicon masters as previously described[41]. Stamps were washed in ethanol and water, then coated with fluorescently labeled streptavidin (10 μg/mL Alexa Fluor 647, ThermoFisher Scientific) for 1 h at room temperature. After PBS washing to remove excess proteins, the stamps were pressed onto 35-mm glass coverslips (#1.5) to transfer the streptavidin. Coverslips were then incubated with the following unbiotinylated proteins to coat the spaces between streptavidin dots: (1) ICAM-spot—10 μg/mL H-2K$^b$-OVA, (2) Antigen-spot – 10 μg/mL ICAM-1, (3) Dual-spot – 5% BSA, (4) ICAM-spot with anti-CD3 background – 10 ug/mL anti-CD3 antibody (Clone 145-2C11, eBioScience), (5) Anti-CD3-spot—10 μg/mL ICAM-1, and (6) control—10 μg/mL unlabeled streptavidin. After 1 h at room temperature, coverslips were washed with PBS three times to wash away uncoated protein and blocked with 5% BSA at room temperature for 1 h before another round of PBS washes. Coverslips were then incubated with the following biotinylated proteins to load the streptavidin: (1) ICAM-spot—2 μg/mL ICAM-1, (2) Antigen-spot—2 μg/mL H-2K$^b$-OVA, (3) Dual-spot—2 μg/mL H-2K$^b$-OVA and 2 μg/mL ICAM-1, (4) ICAM-spot with anti-CD3 background—2 μg/mL ICAM-1, (5) anti-CD3-spot—2 μg/mL anti-CD3 antibody (Clone 145-2C11, eBioScience), and (6) control—2 μg/mL H-2K$^b$-OVA and 2 μg/mL ICAM-1. After 1 h at room temperature, the surfaces were washed into RPMI containing 5% (v/v) fetal calf serum (FCS) and lacking phenol red for imaging. Cells were then added and imaging was performed using either a Leica SP5-inverted confocal laser scanning microscope fitted with 488-, 563-, and 647-nm lasers, or a Leica SP8-inverted confocal laser scanning microscope fitted with a white light laser. In general, samples were imaged every 15 s for 30 min. Additional information about antibodies may be found in Supplementary Table 1.

**Calcium imaging**. CTLs were loaded with 5 μg/mL Fura-2-AM (ThermoFisher Scientific), washed, and then imaged on stimulatory glass surfaces coated with H-2K$^b$-OVA and ICAM-1 as previously described[6]. In all, 340 nm and 380 nm excitation images were acquired every 30 s for 30 min using a 20× objective lens (Olympus).

**MTP surface preparation and imaging**. MTP surfaces were assembled using the following oligos. (1) A21B: /5AmMC6/-CGC ATC TGT GCG GTA TTT CAC TTT-/3Bio/, 2) Quencher strand: /5DBCON/ - TTT GCT GGG CTA CGT GGC GCT CTT- /3BHQ_2/, and 3) Hairpin strand: GTG AAA TAC CGC ACA GAT GCG TTT GTA TAA ATG TTT TCA TTT ATA CTT TAA GAG CGC CAC GTA GCC CAG C. A mixture of oligo A21B (10 nmol) and excess Cy3B-NHS ester or Atto647N-NHS ester (50 μg) in 0.1 M sodium bicarbonate solution was allowed to react at room temperature overnight. The derivatized oligo was then purified by gel filtration and reversed-phase HPLC. Glass coverslips (No. 1.5H, ibidi) were sonicated in MilliQ H$_2$O and ethanol, rinsed in H$_2$O, and then immersed in piranha solution (3:1 sulfuric acid:H$_2$O$_2$) for 30 min to remove organic residues and activate hydroxyl groups on the glass. Subsequently, the cleaned substrates were rinsed with more H$_2$O and ethanol and then transferred into a 200 mL beaker containing 3% APTES in ethanol for 1 h, washed with ethanol, and baked at ~100 °C for 30 min. After cooling, slides were mounted to 6-channel microfluidic cells (Sticky-Slide VI 0.4, ibidi). To each channel, ~50 μL of 10 mg/mL NHS-PEG4-azide in 0.1 M NaHCO$_3$ (pH = 9) was added and incubated for 1 h. The channels were then washed with H$_2$O, blocked with 0.1% BSA in PBS for 30 min, and washed with PBS. ~50 μL of PBS solution was retained inside the channel after washing to prevent drying. Subsequently, the hairpin tension probes were assembled in 1 M NaCl by mixing the Atto647N labeled A21B strand (220 nM), Quencher strand (220 nM), and Hairpin strand (200 nM) in the ratio of 1.1: 1.1:1. The mixture was heat annealed by incubating at 95 °C for 5 min, followed by cooling down to 25 °C over 30 min. ~50 μL of the assembled probe was added to the channels (total volume = ~100 μL) and incubated overnight at room temperature. The following day, unbound DNA probes were removed by PBS wash. Then, 10 μg/mL of streptavidin was incubated in the channels for 45 min at room temperature. The surfaces were cleaned with PBS and incubated with 5 μg/mL of biotinylated pMHC ligand for 45 min at room temperature. After PBS washing, a second DNA tension probe (Cy3B labeled) was assembled and attached as described above, followed by loading with streptavidin and 5 μg/mL of biotinylated ICAM-1. Monomeric ICAM-1 was used for pHluorin-Lamp1 experiments (Fig. 5) and dimeric Fc-ICAM-1 for talin-KO experiments (Figs. 6 and 8). After washing off the unbound ICAM-1 protein, surfaces were rinsed in complete RPMI (no phenol red, supplemented with IL-2) in preparation for imaging with CTLs. MTP imaging was performed on a Nikon Eclipse Ti microscope attached to an electron-multiplying charge-coupled device (EMCCD; Photometrics), an Intensilight epifluorescence source (Nikon), a CFI Apo 100× (NA 1.49) objective lens (Nikon), and a TIRF

launcher with 488-, 561-, and 638-nm laser lines. In general, IRM, 488-, 561-, and 638-nm images were collected every 20 s for 30 min. TIRF illumination was used to image pHluorin-Lamp1 and epifluorescence to image the MTPs.

**Fixed cell immunofluorescence**. OT-1 CTLs were applied to stimulatory glass surfaces coated with streptavidin-AlexaFluor647 followed by biotinylated OVA-H-2K$^b$ (2 μg/mL) and ICAM-1 (2 μg/mL). After 15 min at 37 °C, the cells were fixed by adding 4% paraformaldehyde for 5 min, washed with PBS, and further fixed with methanol for 20 min. Samples were then permeabilized with 0.1% Triton X-100 for 5 min, blocked in PBS solution supplemented with 2% goat serum for 1 h at room temperature, and stained overnight at 4 °C with anti-pericentrin (Abcam, ab4448). Actin was visualized with Alexa 488-labeled phalloidin (ThermoFisher Scientific) using samples fixed with PFA but not methanol. After washing, the appropriate fluorescently labeled secondary antibodies were added for 2 h at room temperature, followed by washing and then imaging using a Leica SP5 confocal laser scanning microscope fitted with 488-, 563-, and 647-nm lasers and a 63× objective lens. Additional information about antibodies may be found in Supplementary Table 1.

**Synapse formation on supported lipid bilayers**. Supported lipid bilayers containing biotinyl cap phosphoethanolamine were prepared as previously described[74]. After loading with streptavidin, the bilayers were incubated with biotinylated OVA-H-2K$^b$ (1 μg/mL) with or without biotinylated ICAM-1 (1 μg/mL). OT-1 CTLs were then applied to the bilayers for 15 min at 37 °C, fixed by adding 4% paraformaldehyde for 5 min, and permeabilized with 0.1% Triton X-100 for 5 min, followed by incubation in PBS solution supplemented with 2% goat serum for 1 h at 37 °C. Cells were then incubated overnight with 0.1 U/ml Alexa Fluor 488-labeled phalloidin dissolved in PBS supplemented with 2% goat serum. Subsequently, samples were washed and imaged by total internal reflection microscopy using a Zeiss Elyra 7 with a 63× objective lens (1.4 NA; Zeiss).

**Chemokinesis**. CTLs were stained for 20 min with Hoechst dye (2 μM final concentration, Thermo Fischer #62249) to facilitate segmentation and tracking. Then, the cells were seeded onto ICAM-1-coated chamber slides (Nunc) and allowed to adhere for 30 min to 1 h before imaging. Three washes in complete media (RPMI with 10% FCS) were performed to remove any nonadherent cells prior to imaging. Fresh complete imaging media (without phenol red) was added, and cells were imaged on a Nikon inverted confocal spinning disk microscope (Ti Eclipse with a Yokogawa CSU-W1, Nikon, Japan). Each video was 30 min in duration, with brightfield and Hoechst channel images taken every 30 s.

**Imaging analysis**. Imaging data were analyzed using SlideBook (3i), Imaris (Bitplane), Excel (Microsoft), Prism (GraphPad), and Python in Jupyter Notebook[75]. Ca$^{2+}$ signaling was quantified by determining the mean Fura-2 ratio for all cells in the imaging field using a mask thresholded on the 340 nm excitation signal. To quantify force exertion in traction force microscopy experiments, custom MATLAB scripts were used to extract pillar displacements from the imaging data, which were then converted into force vectors[40]. To measure the distance between degranulation events and the closest streptavidin spot on micropatterned surfaces, pHluorin-Lamp1 and streptavidin Alexa Fluor 647 signals were converted into Imaris spot constructs using Imaris scripts. The distances between each pHluorin-Lamp1 signal of interest and the closest streptavidin spot within the synaptic boundary of the CTL were then determined using the Imaris "Shortest Distance" function. The expected distance between randomly placed degranulation events and ligand spots was determined in silico. First, the unit cell of the micropattern was modeled as a 5 × 5 μm square with a quarter-circle representing the stamped protein at one corner. Then, the unit square was divided into 1 × 10$^6$ points (evenly sampling of 10-nm increments in both $x$ and $y$), and the Euclidean distance of each point to the stamped protein corner was calculated. The mean distance of this distribution is 3.8266 μm. To quantify the accumulation of CD11a-YFP at microstamped spots of anti-CD3 or ICAM-1, z-projection images (10 × 10 μm squares) centered on each spot were aligned and then averaged over all xy positions. MTP data were analyzed by comparing the mean fluorescence intensity of each MTP within the 2 × 2 μm box centered on a pHluorin-Lamp1 signal of interest with the mean fluorescence intensity of the MTP within the entire IS, defined by threshold masking of IRM images (Fig. 5d). Linescan analysis of ICAM-1-MTP fluorescence at degranulation sites (Fig. 5e) was performed by generating a series of 2-μm linescans bisecting degranulation events of interest. The linescan intensities were aligned around the degranulation, averaged over each pixel, and normalized per linescan. An analogous set of control linescans, collected from parts of the IS lacking degranulation events, were processed in parallel using the same scripts. Centrosome polarization was quantified by measuring the distance between the centrosome and the fluorescent stimulatory surface in Imaris. F-actin polarization was determined starting from cropped volumes encompassing individual cells. In each volume, the mean phalloidin fluorescence of every z-section (0.47 μm per section) was computed starting from the stimulatory surface and moving up. The data were then normalized to the highest value for each cell before the resulting polarization curves were averaged between cells. CTL migration was quantified in Fiji using the TrackMate plugin. Nonadherent cells that floated away

during the videos were excluded from the analysis as they had a larger than 10 μm gap distance between frames. Once tracks were generated, track distance (total distance covered by the cell), instantaneous velocity, and track displacement (the distance from the cell position at the start of the recording) were calculated using an in-house Python script.

**ICAM-1 adhesion**. Flat-bottomed 96-well plates were coated with 10 μg/mL streptavidin in PBS followed by increasing concentrations of biotinylated ICAM-1 in Hepes buffered saline (10 mM HEPES pH 7.5, 150 mM NaCl) with 2% BSA. The total concentration of biotinylated protein during coating was kept at 5 μg/ml by the addition of nonspecific biotinylated protein (H-2D$^b$). OT-1 CTLs were fluorescently labeled with cell trace violet (CTV), resuspended in adhesion buffer (PBS with 0.5% BSA, 2 mM MgCl$_2$, 1 mM CaCl$_2$), and added to ICAM-1-bearing wells in quadruplicate. After a 20 min incubation at 37 °C, wells were washed with warmed adhesion buffer as described[76], and the bound cells were quantified by fluorimetry.

**Immunoblot**. In total, 0.2–1 × 10$^6$ CTLs were lysed using cold cell lysis buffer containing 50 mM Tris-HCl, 0.15 M NaCl, 1 mM EDTA, 1% NP-40, and 0.25% sodium deoxycholate. Suppression of talin 1 was confirmed using an anti-talin 1 antibody (clone 8D4, Abcam). Talin head domain GFP fusion was detected using a polyclonal antibody against GFP (Invitrogen). Actin (clone AC-15, Sigma) or GAPDH (clone D16H11, Cell Signaling Technology) served as loading controls. For signaling assays, serum and IL-2 starved OT-1 CTLs were incubated with streptavidin polystyrene beads (Spherotech) coated with H-2K$^b$-OVA and ICAM-1 at a 1:1 ratio for various times at 37 °C and immediately lysed in 2× cold lysis buffer containing phosphatase inhibitors (1 mM NaF and 0.1 mM Na$_3$VO$_4$) and protease inhibitors (cOmplete mini cocktail, EDTA-free, Roche). Activation of PI3K and MAP kinase signaling was assessed by immunoblot for pAkt (Phospho-Akt (Ser473) Ab; Cell Signaling Technology) and pErk1/2 (Phospho-Thr202/ Tyr204; clone D13.14.4E; Cell Signaling Technology). Additional information about antibodies may be found in Supplementary Table 1. Uncropped images of blots are provided in Supplementary Fig. 12.

**Reporting summary**. Further information on research design is available in the Nature Research Reporting Summary linked to this article.

## Data availability
The raw numbers for charts and graphs are available in the Source Data file whenever possible. Other raw imaging and flow cytometry data relevant to this study are available from the corresponding author upon reasonable request. Unique biological materials, such as plasmids, are also available upon request from the corresponding author. Source data are provided with this paper.

## Code availability
Custom code is publicly available on Github or the Columbia Academic Commons and can be accessed using the following links: https://github.com/msw365/Integrins_degranulation_manuscript_code_Nat_Comm_2022/tree/main; https://github.com/axiezai/degranulation_MTP_fluorescence_average_line_scan_2021; https://doi.org/10.7916/mwk7-e008.

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

## Acknowledgements

We thank L. Stafford and C. Jeronimo for technical support; Y. Romin, E. Chan, M. Tipping, V. Boyko, J. Muller, and the MSKCC Molecular Cytology Core Facility for assistance with imaging; the R.B. Jones lab for assistance with human T cells, and members of the M.H. and L.C.K. laboratories for advice. Supported in part by the NIH (R01-AI087644 to M.H., R01-GM131099 to K.S., R01-AI110593 to L.C.K., and P30-CA008748 to MSKCC), the Ludwig Foundation for Cancer Immunotherapy (M.D.J.), the Schmidt Science Fellows Program (B.Y.W.), and the Cancer Research Institute (B.Y.W.).

## Author contributions

M.S.W., Y. Hu., E.E.S., B.Y.W., K.S., and M.H. designed the experiments. M.S.W., Y. Hu., E.E.S., W.J., E.A.Z., C.S., B.Y.W., and M.H. collected the data. M.S.W., Y. Hu., E.E.S., X.X., M.D.J., B.Y.W., and M.H. analyzed the data. N.H.R., J.H.L., Y. Hong, M.K., and L.C.K. contributed key reagents. M.S.W. and M.H. wrote the paper.

## Competing interests

The authors declare no competing interests.
