## [Peer Review File · Nature Communications]

Mechanically active integrins target lytic secretion at the immune synapse to facilitate cellular cytotoxicityEditorial Note: This manuscript has been previously reviewed at another journal that is not operating a transparent peer review scheme. This document only contains reviewer comments and rebuttal letters for versions considered at Nature Communications.

REVIEWERS' COMMENTS

Reviewer #1 (Remarks to the Author):

The authors have addressed my concerns. In particular, inclusion of the new data with the talin-head construct is an importation piece of data supportive of their main claim.

Reviewer #2 (Remarks to the Author):

In the previous review process, I raised three major concerns and 2 of them have been directly addressed by additional experiments. The one left unaddressed is the physiological relevance of the conclusion that LFA-1 pulling force but not TCR pulling force defines the degranulation domain. The conventional model of degranulation is that the central TCR zone, named cSMAC, serves as the place of degranulation where actin network is depleted to enable granule docking and fusion with PM. LFA-1 molecules cluster at the peripheral of cSMAC, forming a ring-like structure named pSMAC. According to this study, degranulation should preferentially happen at the pSMAC and this model thus is very different from the conventional one. This reviewer still believes that it is important to directly prove this point in a physiologically relevant system other than MTP to revise the conventional model.

Reviewer #4 (Remarks to the Author):

The authors have responded to the comments of Reviewer #3 in a satisfactory manner. One exception is the authors' response to comment #3 and the corresponding new data in Fig. S6, in which they used CRISPR-mediated depletion of talin. The authors need to provide data (e.g., Western blot or imaging) demonstrating proper depletion of talin. In addition, the figure legend should explain the terms "NTCR" and "TInCR".

We thank the reviewers for their constructive comments and suggestions. Below is a point-by-point response, with reviewer comments in blue, responses in black.

Reviewer #2

In the previous review process, I raised three major concerns and 2 of them have been directly addressed by additional experiments. The one left unaddressed is the physiological relevance of the conclusion that LFA-1 pulling force but not TCR pulling force defines the degranulation domain.

The conventional model of degranulation is that the central TCR zone, named cSMAC, serves as the place of degranulation where actin network is depleted to enable granule docking and fusion with PM. LFA-1 molecules cluster at the peripheral of cSMAC, forming a ring-like structure named pSMAC. According to this study, degranulation should preferentially happen at the pSMAC and this model thus is very different from the conventional one. This reviewer still believes that it is important to directly prove this point in a physiologically relevant system other than MTP to revise the conventional model.

Degranulation is canonically thought to occur in a central, F-actin hypodense domain, also called the cSMAC. Over the past decade, however, a number of studies have challenged this canonical model, revealing not only that F-actin is present at the center of the synapse for the duration of its lifetime but also that degranulation events are associated with highly dynamic F-actin remodeling in their immediate vicinity. Additional studies, including our present work, document an important role for integrin engagement in directing cytotoxic secretion. Hence, we favor a mechanistic model in which degranulation requires dynamic and local interplay between ligand bound integrins and F-actin remodeling. We have added a new paragraph to the Discussion section (paragraph #2) that addresses this issue.

Reviewer #4

The authors have responded to the comments of Reviewer #3 in a satisfactory manner. One exception is the authors' response to comment #3 and the corresponding new data in Fig. S6, in which they used CRISPR-mediated depletion of talin. The authors need to provide data (e.g., Western blot or imaging) demonstrating proper depletion of talin. In addition, the figure legend should explain the terms "NtCR" and "TlnCR".

An immunoblot documenting talin depletion is shown in Supplementary Fig. 7a. NtCR and TlnCR are now defined in the figure legends.